# FAME : FACTOR-AWARE MIXTURE-OF-EXPERTS WITH PRETRAINED ENCODER FOR COMBINATORIAL GENERALIZATION

## ABSTRACT

The integration of pretrained encoders with diffusion policies has emerged as a dominant paradigm for visual robotic manipulation. However, it still struggles to generalize across complex environments with varying factors like lighting and surface textures. To address this, we propose FAME, a framework that integrates a factor-aware mixture-of-experts (MoE) with a pretrained encoder to significantly enhance generalization to environmental variations. FAME involves a three-stage training process. (1) policy warmup, where a diffusion policy is trained on data from a standard environment using a frozen encoder. (2) factor-specific adapter training, where we separately train a series of lightweight adapters, inserted between the frozen encoder and the temporally frozen policy, on customized datasets, each focusing on a distinct environmental variation. (3) joint fine-tuning, where we simultaneously train a centric router and the warmed policy on a mixed dataset to handle multiple factors at once. We say FAME is "factor-aware" because the central router organizes the frozen factor-specific adapters as a MoE, allowing for combinatorial generalization for multiple factors. Evaluations on the Meta-World benchmark with various environmental factors show that our proposed FAME significantly outperforms existing diffusion policy baselines. Furthermore, FAME demonstrates remarkable scaling properties as the number of demonstrations increases. We believe our FAME provides an effective solution for achieving combinatorial generalization in visual robotic control tasks.

## 1 INTRODUCTION

The adoption of Diffusion Policies (DP) Chi et al. (2023a) has become a well-established consensus in visual robotic manipulation, owing to their powerful fitting capabilities for complex, high-dimensional tasks. This has led to the prevailing approach of integrating DP with various pretrained visual encoders, which provides rich, transferable feature representations without requiring extensive task-specific data. Nevertheless, the architecture and adaptation strategies of these encoders still present a substantial design space with considerable room for exploration (Nair et al., 2022).

which provides rich, transferable feature representations without requiring extensive task-specific data. Representative encoders includes DINOv2 (Oquab et al., 2023), CLIP (Radford et al., 2021) and R3M (Nair et al., 2022).

Despite these advancements, current methods still struggle to generalize across complex environments with varying factors such as lighting, surface textures, or camera viewpoints. If mastering each factor requires additional data of size $N$, then simultaneously handling $K$ independent factors could imply a considerable data complexity of $N^K$, which becomes prohibitively expensive in practice.

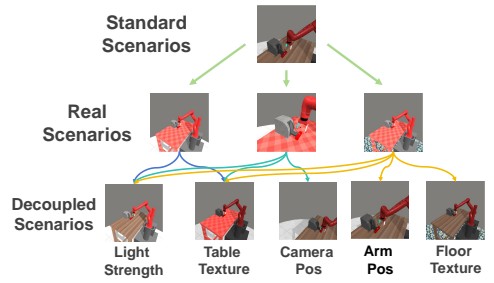

Figure 1: Decomposition of environmental variations into independent factors.

Fortunately, many factors in the physical world vary independently. This observation motivates a divide-and-conquer strategy: by disentangling and separately learning each factor, we can potentially reduce the data requirement from exponential to approximately linear, i.e., $N \times K$. As illustrated in Figure 1, real-world environmental changes can be decomposed into discrete and independent factors. Explicitly modeling these variations enables more systematic and combinatorial adaptation to a majority of common conditions.

Furthermore, directly fine-tuning large pre-trained encoders remains challenging: it is computationally expensive, prone to overfitting, and often results in catastrophic forgetting of pre-trained knowledge. To overcome these limitations, we propose a structured approach that factorizes environmental variations, enabling efficient and scalable combinatorial generalization in complex visual manipulation tasks.

In this paper, we introduce **FAME** (Factor-Aware Mixture-of-Experts with Pretrained Encoder), a novel framework that enhances the generalization capability of diffusion policies through factor-aware adaptation. FAME incorporates a Mixture-of-Experts (MoE) architecture that dynamically combines lightweight, factor-specific adapters, each dedicated to a specific environmental variation. The training process consists of three stages: (1) **Policy warm-up**: A diffusion policy is first trained using a frozen pretrained encoder on data from a standard environment. (2) **Factor-specific adapter training**: Lightweight adapters are inserted between the encoder and the policy network and trained separately on specialized datasets, each targeting a distinct environmental factor. (3)**Joint fine-tuning**: A central router is trained along with the policy on a mixed dataset to combine adapters dynamically and achieve combinatorial generalization.

Extensive experiments on the Meta-World benchmark demonstrate that FAME significantly outperforms existing diffusion policy baselines in environments with diverse factors. The framework also exhibits remarkable scaling behavior with increasing demonstration data and maintains strong performance under single-factor variations.

Our contributions are summarized as follows:

- **FAME Framework**: We propose FAME, a factor-aware framework that integrates a Mixture-of-Experts(MoE) architecture with a frozen pretrained encoder to handle compound environmental variations in visual robotic manipulation.

- **Three-Phase Factor-Aware Training**: We design a three-stage training procedure that includes policy warm-up, factor-specific adapter training, and joint fine-tuning with a router, enabling efficient and scalable adaptation.

- **Experiment Validation**: We conduct extensive experiments showing that FAME achieves superior generalization performance compared to strong baselines and demonstrates excellent scalability with respect to demonstration data.

## 2 RELATED WORK

**Diffusion policy and robotic manipulation.**  Diffusion models, which progressively transform random noise into structured data samples, have demonstrated remarkable success in high-fidelity image generation, as exemplified by DDPM (Ho et al., 2020; Song & Ermon, 2020). Owing to their strong representational power, such models are increasingly being adopted in robotics. For instance, they have been applied in reinforcement learning (Wang et al., 2024; Li et al., 2025; Gu et al., 2025; Sheng et al., 2025), and in imitation learning (Chi et al., 2023b; Huang et al., 2025; Tie et al., 2025). In this work, we focus on leveraging diffusion models for robotic manipulation under complex generalization scenarios. We investigate how diffusion-based policies, formulated as conditional diffusion models, can be improved through architectural modifications to enhance the generalization capability of robotic policy learning.

**Pre-trained visual encoders.**  In the realm of computer vision, several prominent pre-trained visual encoders have emerged as powerful feature extractors, including Vision Transformer (ViT) (Dosovitskiy et al., 2021), DINOv2 (Oquab et al., 2023), and CLIP (Radford et al., 2021). Among these, DINOv2—a robust visual encoder based on self-supervised learning—has been extensively applied in embodied motion vision due to its strong representation capabilities. These general-purpose encoders

have subsequently inspired and facilitated the development of specialized visual encoders within the field of robotic policy learning. Notable contributions include MVP (Xiao et al., 2022), R3M (Nair et al., 2022), VIP (Ma et al., 2022), and VC-1 (Majumdar et al., 2024), which leverage large-scale pre-training to provide effective visual representations that serve as valuable prior knowledge for training robot policies. In this paper, we employ the pre-trained visual representations from DINOv2 (Oquab et al., 2023) and our framework is compatible to any other encoders.

**Parameter-efficient fine-tuning.** Despite the strong representational capabilities of pre-trained visual encoders, their limited adaptability to environmental variations poses a significant challenge for robotic manipulation. To address this issue, we draw inspiration from Parameter-Efficient Fine-Tuning (PEFT) methods developed in natural language processing. Instead of full fine-tuning that updates all parameters, these approaches introduce small trainable modules into frozen pre-trained backbones, preserving the original representations while enabling task-specific adaptation. Seminal work in this area includes Adapter modules (Houlsby et al., 2019) and Low-Rank Adaptation (LoRA) (Hu et al., 2021), alongside other techniques like Prompt Tuning (Lester et al., 2021) and Prefix Tuning (Li & Liang, 2021). These methods have demonstrated remarkable success in adapting large language models with minimal computational overhead. Our work extends this parameter-efficient paradigm to visual representation learning for robotic manipulation, developing factor-specific adapters that maintain the benefits of large-scale pre-trained visual encoders while enabling efficient adaptation to diverse environmental conditions.

**Mixture-of-Experts (MoE) frameworks.** The MoE architecture provides an effective mechanism for dynamically integrating multiple specialized modules. Originally introduced by (Shazeer et al., 2017), MoE enables scalable neural networks by selectively routing inputs to specialized "expert" sub-networks. This approach has demonstrated remarkable success in large language models, including the Switch Transformer (Fedus et al., 2021) and Mixtral 8x7B (Jiang et al., 2023). Beyond natural language processing, MoE has been effectively applied in autonomous driving for multi-modal perception and adaptive planning (Liu et al., 2022; Wang et al., 2023), as well as in robotics for acquiring diverse manipulation skills (Fu et al., 2022; Gupta et al., 2023). Our work innovatively combines the concepts of parameter-efficient adaptation and mixture-of-experts by developing a FAME framework that dynamically integrates factor-specific adapters. This approach allows the system to selectively combine specialized adapters based on the current environmental context, effectively addressing the challenge of combinatorial generalization in robotic manipulation scenarios.

## 3 METHOD

In this section, we introduce the core methodology of the FAME framework. This framework addresses the challenge of diverse environmental variations in robotic manipulation by combining a three-phase training approach with a dynamic MoE mechanism and knowledge distillation.

### 3.1 OVERVIEW OF FAME FRAMEWORK

The framework of our FAME is illustrated in Figure 2, where the training process is summarized using color-coded arrows: green arrows denote policy warm-up (Phase 1 in Section 3.2), in which a diffusion policy is trained using a frozen pretrained encoder on data from a standard environment; gray arrows represent factor-specific adapter training (Phase 2 in Section 3.3), where lightweight adapters are inserted and trained separately on specialized datasets, each targeting a distinct environmental factor; and blue arrows correspond to joint fine-tuning (Phase 3 in Section 3.4), during which a central router is trained along with the policy on a mixed dataset to combine adapters dynamically.

Before detailing the model architecture and training procedures in the following subsections, we first introduce the three types of datasets used across different stages of the training framework:

(1) **Standard Dataset** ($\mathcal{D}_0$): Data collected in the standard manipulation task environment.

(2) **Gen Dataset** ($\mathcal{D}_k$): Data collected under environments where only the $k$-th factor (e.g., light strength) is varied relative to the standard setup, for each $k \in 1, \dots, K$.

(3) **Mix Gen Dataset** ($\mathcal{D}_{\textbf{multi}}$): Data collected under environments where any subset of $i$ factors vary simultaneously, with $i \in \{2, 3, 4, K\}$.

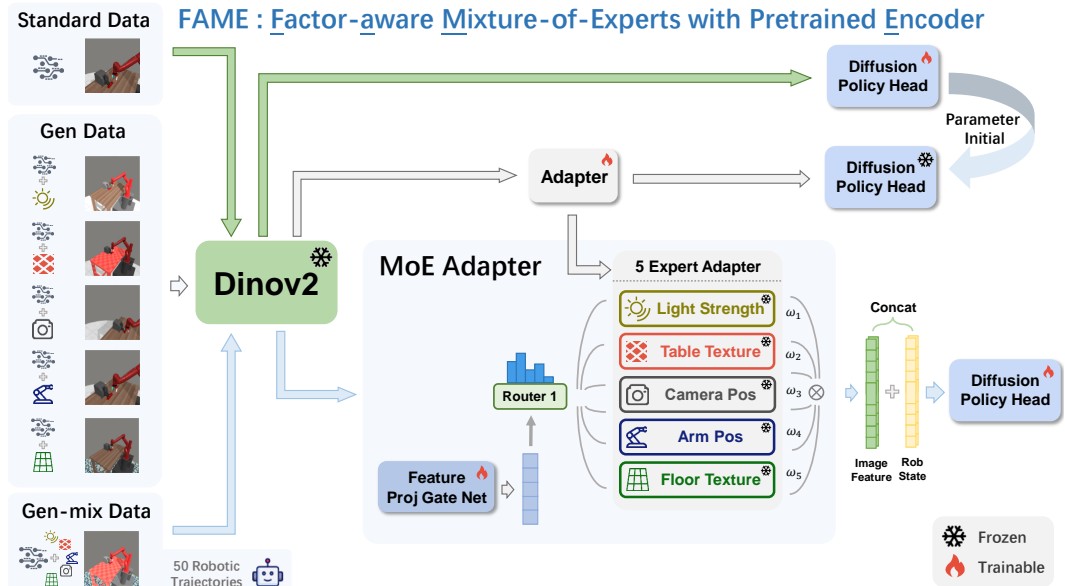

Figure 2: FAME framework: (1) Policy warm-up: The standard DP framework serves as the baseline policy training; (2) Factor-specific adapter training: Multiple adapters are trained on a frozen DINOv2 backbone to handle individual variations (e.g., lighting, texture); (3) Joint fine-tuning: A gating network dynamically combines adapter outputs via Mixture-of-Experts.

## 3.2 PHASE 1: POLICY WARM-UP

The first phase aims to learn a base policy that performs well under standard environmental conditions.

We adopt the two-stage end-to-end diffusion policy (DP) architecture as the backbone of our framework. The first stage employs a visual backbone based on a frozen pre-trained DINOv2 (Oquab et al., 2023) model to leverage its powerful representation capabilities. The second stage consists of a diffusion policy head which is trained from scratch. Training uses standard task data $\mathcal{D}_0$ from Section 3.1 without environmental variations. Given input observation $\mathbf{o}_t^0$ (where the top-right label "0" represents the dataset to which $\mathbf{o}_t$ belongs), the visual backbone extracts features $\mathbf{f}_v = \mathcal{H}^{\text{frozen}}_{\text{DINOv2}}(\mathbf{o}_t^0)$, and the DP head generates actions $\mathbf{a}_t = \mathcal{H}_{\text{DP}}(\mathbf{f}_v)$. The training objective is to minimize the loss function:

$$\min_{\theta_{\mathcal{H}_{\text{DP}}}} \mathcal{L}_{\text{DP}}(\mathcal{D}_0; \theta_{\mathcal{H}_{\text{DP}}}, \mathcal{H}^{\text{frozen}}_{\text{DINOv2}}), \tag{1}$$

where $\theta_{\mathcal{H}_{\text{DP}}}$ denotes the parameters of the diffusion policy. This phase establishes a strong baseline policy that performs well under standard environmental conditions.

## 3.3 PHASE 2: FACTOR-SPECIFIC ADAPTER TRAINING

In the second phase, we train specialized adapter networks for each environmental factor while keeping both the visual backbone and the DP head frozen.

For each environmental factor $k \in \{1, \ldots, K\}$, we introduce a trainable adapter network $\mathcal{A}_k$ between the frozen DINOv2 and the frozen DP head obtained from Phase 1. The visual features $\mathbf{f}'_v$ are first extracted by the frozen DINOv2 model as $\mathbf{f}'_v = \mathcal{H}^{\text{frozen}}_{\text{DINOv2}}(\mathbf{o}_t^k)$, where $\mathbf{o}_t^k$ denotes the input observations from dataset $\mathcal{D}_k$ in Section 3.1. The adapter network $\mathcal{A}_k$ then transforms these features into adapted visual features $\mathbf{f}_v^k = \mathcal{A}_k(\mathbf{f}'_v)$, which are passed through the frozen DP head to obtain the output $\mathbf{a}_t = \mathcal{H}^{\text{frozen}}_{\text{DP}}(\mathbf{f}_v^k)$. The training objective for the adapter $\mathcal{A}_k$ is to minimize the loss function $\mathcal{L}_{\text{DP}}$ with respect to the adapter's parameters $\theta_{\mathcal{A}_k}$, while keeping the DINOv2 and DP head models frozen:

$$\min_{\theta_{\mathcal{A}_k}} \mathcal{L}_{\text{DP}}(\mathcal{D}_k; \theta_{\mathcal{A}_k}, \mathcal{H}^{\text{frozen}}_{\text{DINOv2}}, \mathcal{H}^{\text{frozen}}_{\text{DP}}), k \in \{1, \ldots, K\}. \tag{2}$$

This formulation ensures that each adapter $\mathcal{A}_k$ learns to adapt the visual features specifically for the variations present in dataset $\mathcal{D}_k$, effectively specializing in handling a particular environmental factor while maintaining the base policy's core functionality.

### 3.4 PHASE 3: JOINT FINE-TUNING

The final phase integrates the specialized adapter networks through a MoE architecture, enabling dynamic combination of expert representations based on input conditions. The gating mechanism learns to identify which environmental factors are present in the input and appropriately weights the corresponding adapters. The MoE layer comprises two components:

1. **Gating network** $\mathcal{G}$: This network learns to compute adapter weights $\mathbf{w} = [w_1, \ldots, w_k, \ldots, w_K]$ from the visual features $\mathbf{f}_v'$. The gating network essentially acts as a router, determining the contribution of each expert based on the input characteristics.

2. **Adapter bank**: This include pre-trained factor-specific adapter networks $\mathcal{A}_k$ for $k \in \{1, \ldots, K\}$ in the Phase 2 in Section 3.3, which remain frozen during the MoE training process. These adapters serve as specialized experts, each proficient in handling a specific environmental variation.

The final visual representation is obtained by combining the outputs of the adapter networks via a weighted summation:

$$\mathbf{f}_v^{\text{MoE}} = \sum_{k=1}^{K} \underbrace{\text{Softmax}\left(\mathcal{G}(\mathbf{f}_v')\right)}_{w_k} \cdot \underbrace{\mathcal{A}_k(\mathbf{f}_v')}_{\mathbf{f}_v^k} \tag{3}$$

This combined visual representation $\mathbf{f}_v^{\text{MoE}}$ is then passed through the DP head to produce the final output: $\mathbf{a}_t = \mathcal{H}_{\text{DP}}(\mathbf{f}_v^{\text{MoE}})$.

**Training procedure.** During training, we utilize multi-factor variation data $\mathcal{D}_{\text{multi}}$ from Section 3.1 to optimize only the gating Network $\mathcal{G}$ and a new DP head, while keeping the visual backbone and all adapter networks frozen. This training strategy allows the gating network to learn effective combination strategies while preventing catastrophic forgetting of the specialized adapter capabilities. The specific training objective is

$$\min_{\theta_\mathcal{G}, \theta_{\mathcal{H}_{\text{DP}}}} \mathcal{L}_{\text{DP}}\left(\mathcal{D}_{\text{multi}}; \theta_\mathcal{G}, \theta_{\mathcal{H}_{\text{DP}}}, \mathcal{H}_{\text{DINOv2}}^{\text{frozen}}, \mathcal{A}_k^{\text{frozen}}\right), \quad k \in \{1, \ldots, K\} \tag{4}$$

Our framework enables the agent to dynamically adapt to complex environmental conditions by intelligently combining the specialized knowledge of multiple experts, resulting in robust performance across diverse scenarios.

## 4 EXPERIMENT

### 4.1 MAIN EXPERIMENT

**Meta-World benchmark.** Meta-World benchmark (Yu et al., 2020) is a widely recognized platform for robotic manipulation that provides a diverse set of tasks simulating real-world scenarios. We choose a representative subset of **9 tasks** from this benchmark to conduct experiments. Detailed descriptions of these tasks can be found in Appendix A.

**Environment customization.** Meta-World provides only the standard environment interface without variations. To enable our research on generalization, we develop `MetaWorldEnvFactor`, a lightweight wrapper class that can be directly nested on top of the original `MetaWorldEnv`. We implement 5 independent factor variations (object size, color, lighting, friction, and camera pose) and can arbitrarily compose them to customize environments with diverse factor combinations. Further implementation details are given in the Appendix B.

**Traning dataset.** We use Meta-World's built-in policies to construct dataset. By iterating the inference-execution loop until success, high-quality expert trajectories (image-state-action sequences) are collected as the Standard Dataset ($\mathcal{D}_0$). With the help of `MetaWorldEnvFactor`, we can further construct the Gen Dataset ($\mathcal{D}_k$) and Mix Gen Dataset ($\mathcal{D}_{\text{multi}}$). Each dataset contains 50 successful demonstrations.

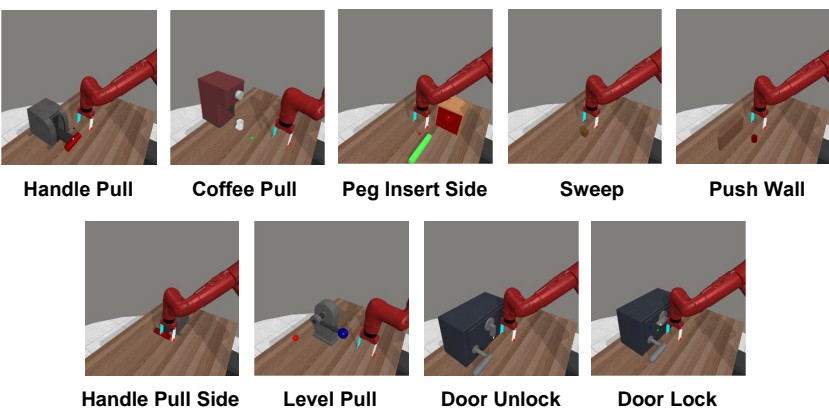

Figure 3: The 9 tasks from Meta-World serving as our experimental benchmark.

**Training setup.** Detailed training hyper-parameters are provided in the Appendix C.

**Evaluation setting.** To thoroughly evaluate the policy robustness, we build 5 test environments for each task. Take Hand-Pull as an example as illustrated in Figure 4, the 5 test environments exhibit progressively increasing complexity, ranging from single-factor variations to the most challenging scenario with all five factors simultaneously involved. Evaluation is performed every 200 epochs, resulting in 10 evaluations over the entire training run of 2000 epochs. In each evaluation round, the model is assessed in all 5 test environments ($i = 1, 2, 3, 4, 5$), yielding 5 individual results. The average of these 5 results is then taken as the evaluation outcome for that particular round. More details regarding the evaluation settings will be provided in the Appendix D.

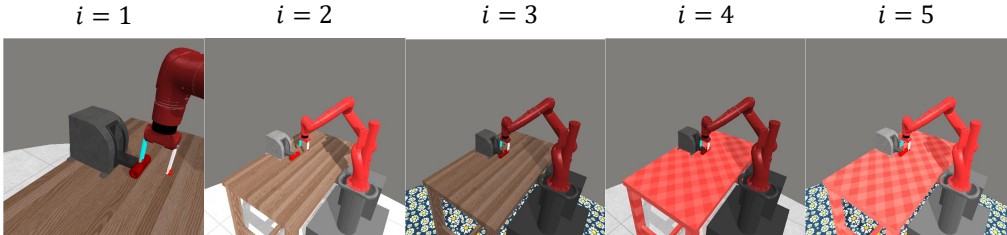

Figure 4: **Visualization of the five evaluation environments with progressively increasing factors.** From left to right: environments with 1 to 5 factors simultaneously varied, demonstrating the increasing complexity of environmental perturbations used for evaluation.

**Baselines.** We consider several well-known method in visual robotic manipulation, including DP with ResNet (He et al., 2016), DP with DINOv2 (Oquab et al., 2023) (a ViT-based encoder that learns high-quality visual representations via self-supervised pre-training on large-scale unlabeled image data), DP with CLIP (Radford et al., 2021) (a vision-language model trained on a massive web-scale dataset of image-text pairs), and DP with R3M (Nair et al., 2022) (self-supervised pre-training on large-scale human video data). Our FAME-DP also employs DP as the downstream controller, while the major difference is that we design a factor-aware MoE to collaborate with DINO encoder for better combinatorial generalization capability.

**Main results.** For each task, we run 3 random seeds and each evaluation result is the average outcome across 5 test environments with 1 to 5 varying factors ($i = 1, 2, 3, 4, 5$). The numerical results of all algorithms are summarized in Table 1 and the curves are shown in Figure 5. Our approach consistently achieves the highest performance, with an average success rate of **54.15%** across all environmental settings, surpassing the second-best method by 34% over 9 tasks. Notably, on challenging tasks such as **Door Lock**, **Handle Pull Side**, and **Peg Insert Side**, our method outperforms all baselines by a large margin—achieving 59.33%, 37.67%, and 60.33% respectively. Furthermore, FAME excels in

tasks like **Door Unlock** and **Lever Pull**, reaching success rates of $93.67\%$ and $84.00\%$, significantly higher than other methods. These results affirm the strong generalization capability of FAME when faced with diverse and unseen environmental variations. Detailed curves for each environment ($i = 1$ to 5) are provided in Appendix E.

Table 1: **Average final success rate.** We report the mean $\pm$ one standard deviation over three random seeds of the evaluation results obtained at the $2000^{\text{th}}$ epoch.

| Alg \ Task | Coffee Pull | Door Lock | Push Wall | Sweep | Lever Pull |
|---|---|---|---|---|---|
| ResNet-DP | $22.67 \pm 3.86$ | $32.33 \pm 6.24$ | $17.67 \pm 3.40$ | $11.67 \pm 1.70$ | $26.33 \pm 17.75$ |
| R3M-DP | $0.33 \pm 0.47$ | $14.67 \pm 3.30$ | $0.00 \pm 0.00$ | $0.33 \pm 0.47$ | $0.00 \pm 0.00$ |
| CLIP-DP | $18.33 \pm 5.44$ | $45.67 \pm 1.25$ | $7.67 \pm 3.86$ | $14.67 \pm 2.62$ | $13.67 \pm 4.19$ |
| DINO-DP | $20.00 \pm 5.72$ | $29.67 \pm 4.71$ | $19.00 \pm 5.35$ | $13.00 \pm 4.08$ | $23.00 \pm 7.35$ |
| FAME (Ours) | $\mathbf{28.00 \pm 0.82}$ | $\mathbf{59.33 \pm 3.68}$ | $\mathbf{42.00 \pm 6.16}$ | $\mathbf{25.33 \pm 6.65}$ | $\mathbf{84.00 \pm 4.97}$ |

| Alg \ Task | Door Unlock | Handle Pull | Handle Pull Side | Peg Insert Side | Average |
|---|---|---|---|---|---|
| ResNet-DP | $40.33 \pm 8.18$ | $12.33 \pm 1.25$ | $7.67 \pm 2.05$ | $10.00 \pm 2.83$ | $20.11$ |
| R3M-DP | $29.33 \pm 11.56$ | $26.00 \pm 1.63$ | $0.00 \pm 0.00$ | $0.00 \pm 0.00$ | $7.85$ |
| CLIP-DP | $54.67 \pm 3.30$ | $0.67 \pm 0.47$ | $1.00 \pm 0.00$ | $3.67 \pm 1.70$ | $17.78$ |
| DINO-DP | $36.00 \pm 3.27$ | $20.00 \pm 3.27$ | $9.67 \pm 6.85$ | $6.67 \pm 1.25$ | $19.67$ |
| FAME (Ours) | $\mathbf{93.67 \pm 3.68}$ | $\mathbf{57.00 \pm 5.89}$ | $\mathbf{37.67 \pm 3.30}$ | $\mathbf{60.33 \pm 5.31}$ | $\mathbf{54.15}$ |

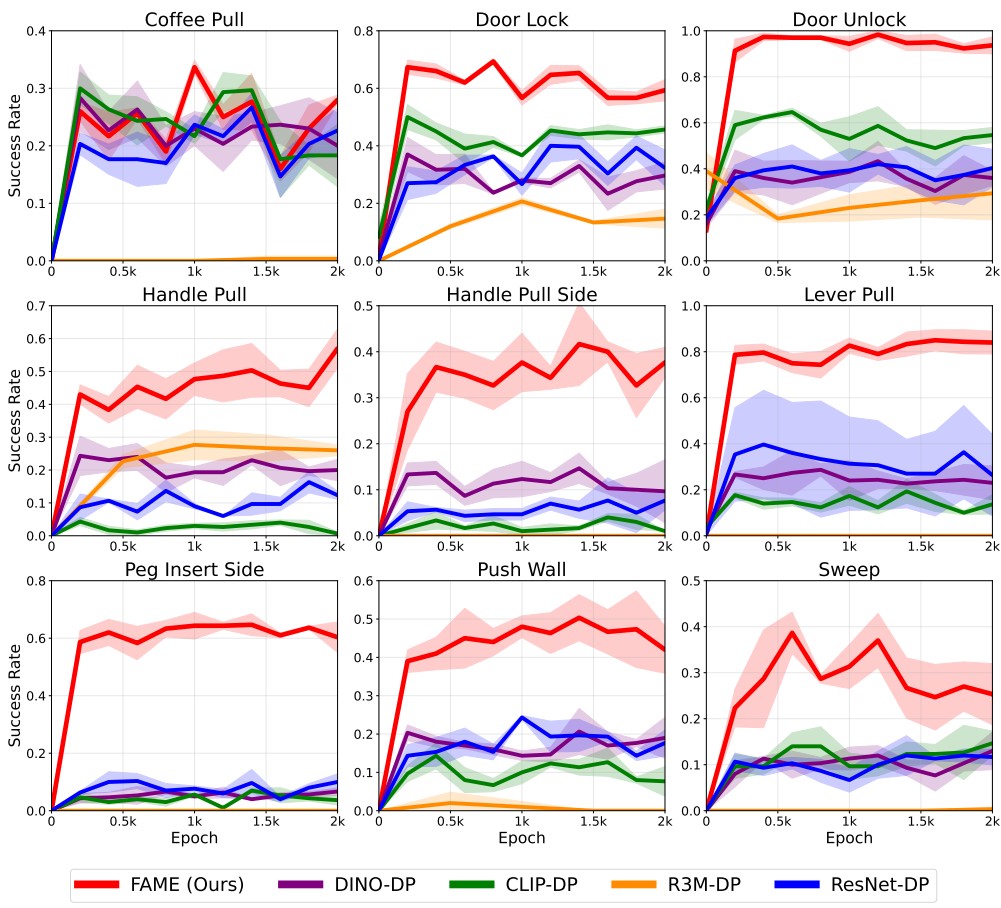

Figure 5: **Training curves on benchmarks.** The solid lines correspond to the mean and shaded regions correspond to one standard deviation over three runs. Each evaluation result is averaged across five environments with $i = 1$, $i = 2$, $i = 3$, $i = 4$, and $i = 5$ varying factors.

## 4.2 Ablation Study

To investigate the core properties of our FAME framework, we conducted a detailed ablation study on the **Handle-Pull** task.

**(1) Scaling property with data increasing.** We evaluated the scaling effects of our FAME framework by training on varying dataset sizes (1, 5, 10, 20, 50, and 100 demonstrations), using the same Mix Gen Dataset ($\mathcal{D}_{\text{multi}}$, $i = 5$) as in the main experiments. As shown in Figure 6, our algorithm consistently outperformed baselines across all scales. The results reveal a strong scaling behavior, with performance improving significantly as data volume increases. This demonstrates that our framework effectively leverages larger datasets to enhance generalization, a key advantage that highlights the effectiveness of combining a pre-trained encoder with a dynamic MoE structure.

**(2) Performance considering only single factor variation at a time.** While our main experiments showed strong performance on multi-factor variations, we also evaluated our FAME framework's ability to handle single-factor changes. For this, we trained and evaluated the model using only the Gen Dataset ($\mathcal{D}_k$), where each environment contained a single varying factor. As shown in Figure 7, our FAME algorithm maintains strong performance across all five individual factors. This demonstrates the framework's robust adaptability, proving it is highly effective at addressing both single-factor and multiple-factors environmental challenges.

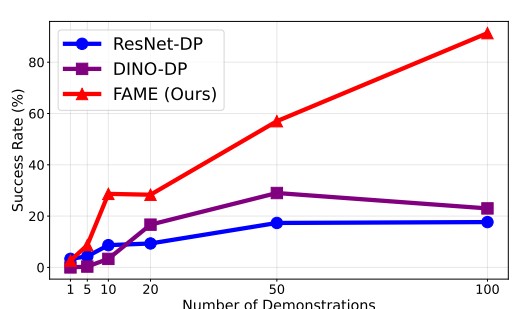

**(3) Dataset sensitivity in the final phase.** To test the robustness of our FAME framework, we replaced the Mix Gen Dataset ($\mathcal{D}_{\text{multi}}$, $i = 5$) used in the main experiments with the Standard Dataset ($\mathcal{D}_0$) in the final phase, using 50 demonstrations per task while keeping all other experimental settings unchanged. As shown in Table 2, both the baseline DP and DINOv2 models suffered a significant performance drop, with DP decreasing by 55.5% and DINOv2 by 39%.

Figure 6: **Scaling performance with increasing demonstration data.** Evaluation of FAME and baselines trained on the Mix Gen Dataset ($\mathcal{D}_{\text{multi}}$) with varying numbers of demonstrations.

In contrast, our FAME model was only minimally affected, maintaining high performance nearly identical to that achieved when trained on explicit generalization data. These results demonstrate that the final phase of our FAME is not sensitive to the dataset diversity and maintains a strong performance. We argue that this is because our factor-specific adapters have learned the essential capability to handle the corresponding variations, and the central router in an MoE exhibits combinatorial generalization, allowing it to handle diverse environmental variations.

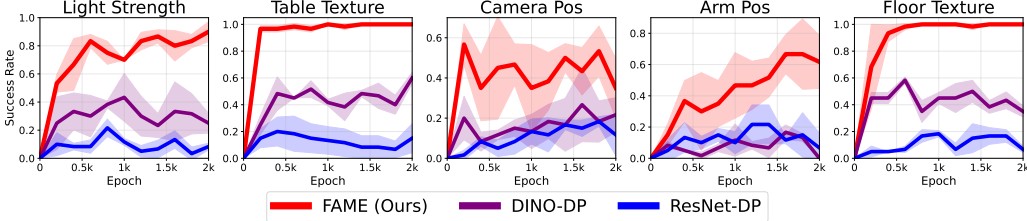

Figure 7: Evaluation on environments containing only one varying factor at a time (Gen Dataset $\mathcal{D}_k$).

Table 2: **Performance comparison using different datasets in the final phase.**

| Dataset | ResNet-DP | DINO-DP | FAME(ours) |
|---|---|---|---|
| $\mathcal{D}_{\text{multi}}$ | $17.3 \pm 2.5$ | $29.0 \pm 0.8$ | $\mathbf{57.0 \pm 5.9}$ |
| $\mathcal{D}_0$ | $7.7 \pm 8.2$ ($\downarrow 55.5\%$) | $17.7 \pm 3.9$ ($\downarrow 39.0\%$) | $\mathbf{56.3 \pm 1.7}$ (Nearly same) |

### 4.3 Zero-shot Cross-Task Generalization of the Gating Network in FAME

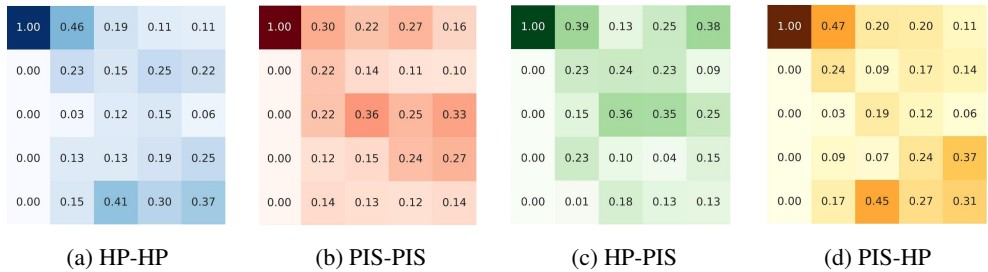

(a) HP-HP      (b) PIS-PIS      (c) HP-PIS      (d) PIS-HP

Figure 8: **Cross-task generalization of the gating network in FAME.** Heatmap visualizations of gating activations on the Handle Pull (HP) and Peg Insert Side (PIS) tasks, demonstrating zero-shot cross-task adaptation without fine-tuning.

To better understand the FAME architecture, this subsection provides a dedicated explanation of the working mechanism of the gating network within FAME. We choose two tasks: **Handle Pull** and **Peg Insert Side**. The gating network is trained using the Gen Dataset ($\mathcal{D}k$) and the Mix Gen Dataset ($\mathcal{D}$multi with $i = 2, 3, 4, 5$). After training, we feed the observations from the same task or the other task into the model, and then visualize the activation values output by the gating network as heatmaps, as shown in Figure 8 (we consider 2 tasks so there are $2 \times 2 = 4$ visualizations). In each subfigure, the horizontal axis represents the number of varying factors($i$) in the training data, ranging from 1 to 5, while the vertical axis indicates the activation value corresponding to each expert adapter network.

As shown in the first two Figure 8a and 8b, when the number of varying factors is small, the gating network tends to focus more on certain specific adapters. As the number of factor variations increases, the activations become more dispersed, reflecting the model's adaptive allocation of experts to handle growing complexity. Notably, as shown in the last two Figure 8c and 8d, we also observe that the gate trained on the **Handle Pull** task can be directly and effectively transferred to the **Peg Insert Side** task in a zero-shot manner. This cross-task generalization capability suggests that the gating network learns a high-level, task-agnostic representation of visual factors, rather than overfitting to task-specific cues. This further demonstrates the effectiveness of combining adapter network fine-tuning with the MoE architecture.

## 5 Conclusion

In this work, we proposed FAME, a novel framework that integrates a Mixture-of-Experts architecture with a frozen pre-trained visual encoder to significantly enhance the combinatorial generalization capability under diverse and complex environmental variations. By training lightweight, factor-specific adapters and combining them dynamically through a gating network, FAME effectively handles both isolated and compounded domain shifts, such as changes in lighting, texture, and camera perspective, without compromising the representation power of the underlying pre-trained backbone. Extensive experiments on a diverse set of Meta-World manipulation tasks demonstrate that FAME consistently outperforms strong baselines, including methods built on pre-trained features (DINOv2, CLIP, R3M) and the standard ResNet diffusion policy. The framework exhibits remarkable scalability with increasing data, strong adaptation to both single-factor and multi-factor variations, and substantial cross-task generalization ability, confirming that the learned representations are both transferable and factor-aware. We believe this work opens up a new direction for training practically useful robots with the enhanced combinatorial generalization capability.

For future work, we plan to explore: (1) extending FAME to a broader set of environmental factors and real-world robotic applications; (2) incorporating reinforcement learning or online fine-tuning to enable continual adaptation in non-stationary settings; and (3) investigating more efficient and interpretable gating mechanisms for real-time policy execution. We believe that the combination of pre-trained encoders with dynamic, factor-wise specialization offers a promising pathway toward more general and deployable robot learning systems.

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

## A    META-WORLD TASK INTRODUCTION

We conduct simulation experiments on 9 tasks selected from the *Meta-World* benchmark (Yu et al., 2020), with brief descriptions as follows:

- **Coffee Pull**: Grasp and pull a mug out of a coffee machine.
- **Door Lock**: Lock a door by rotating the lock clockwise.
- **Door Unlock**: Unlock a door by rotating the lock counter-clockwise.
- **Handle Pull**: Pull a handle upward.
- **Handle Pull Side**: Pull a handle upward sideways.
- **Lever Pull**: Pull a lever down by 90 degrees.
- **Peg Insert Side**: Insert a peg sideways into a target hole.
- **Push Wall**: Bypass a wall and push a puck to a goal.
- **Sweep**: Sweep a puck off the table.

## B    META-WORLD FACTOR WRAPPER

We develop `MetaWorldEnvFactor`, a comprehensive wrapper class that extends the standard Meta-World environment interface to support multi-factorial control and rich sensory observations. This wrapper enables independent manipulation of five distinct environmental factors while maintaining compatibility with the original Meta-World API.

### B.1    CLASS OVERVIEW

The `MetaWorldEnvFactor` class is built upon the OpenAI Gym interface and provides a unified framework for controlling environmental variations in Meta-World tasks. Key features of this wrapper include:

- **Multi-factor control:** Independent manipulation of five environmental factors through a binary encoding system
- **Backward compatibility:** Maintains full compatibility with the original Meta-World environment API
- **Rich observation space:** Provides RGB images, agent proprioception, and full environment state

The class is initialized with parameters specifying the task name, observation configuration, and factor activation pattern:

```
class MetaWorldEnvFactor(gym.Env):
    def __init__(self, task_name, device="cuda:0",
                 seed=None,
                 factors=None):
```

### B.2    INITIALIZATION PROCESS

During initialization, the wrapper performs several key operations:

1. **Environment setup:** Loads the appropriate Meta-World environment, ensuring it uses the goal-observable v2 variant

```
if '-v2' not in task_name:
    task_name = task_name + '-v2-goal-observable'
self.env = metaworld.envs.ALL_V2_ENVIRONMENTS_GOAL_OBSERVABLE[task_name]()
```

2. **Factor parsing:** Interprets the 5-character binary string to determine which factors to apply

```
factors = str(factors)
if factors is not None:
    assert len(factors) == 5
    factors = [bool(int(x)) for x in factors]
    # Set individual factor flags based on binary string
```

3. **Factor application:** Applies the requested environmental modifications in sequence

4. **Camera configuration:** Sets up the default camera position with optional randomization

5. **Observation space definition:** Configures the rich observation space including multiple sensory modalities

## B.3 FACTOR IMPLEMENTATION DETAILS

### B.3.1 LIGHTING VARIATION

The lighting factor modifies both ambient and diffuse lighting properties in the MuJoCo model:

- **Range:** RGB values are sampled uniformly from [0.05, 0.95] for all three channels

- **Implementation:** Direct modification of the XML model's headlight properties using regular expressions

- **Code:**

```
def change_light(env, diffuse_range=(0.05, 0.95), seed=None):
    if seed is not None:
        np.random.seed(seed)
    light = np.full((3, ), np.random.uniform(*diffuse_range))
    ambient = light
    ambient_str = ' '.join([f"{x:.3f}" for x in ambient])
    diffuse = light
    diffuse_str = ' '.join([f"{x:.3f}" for x in diffuse])
```

### B.3.2 TABLE TEXTURE VARIATION

The table texture factor replaces the default table texture with randomly selected alternatives:

- **Source:** PNG files from a figure batch

- **Implementation:** XML texture reference modification for texture named "T_table"

- **Error handling:** 10 attempts with different random textures to ensure successful loading

### B.3.3 CAMERA POSITION VARIATION

The camera position factor modifies the viewpoint from which observations are captured:

- **Default position:** [0.6, 0.295, 0.8]

- **Variation range:**
    - x-axis: [0.5, 0.7]
    - y-axis: [0.2, 0.4]
    - z-axis: [0.7, 0.9]

- **Implementation:** Direct modification of `env.sim.model.cam_pos[2]`

### B.3.4 AGENT INITIAL POSITION VARIATION

The agent position factor introduces noise to the initial end-effector position:

- **Base position:** Original mocap position
- **Noise:** Uniform distribution with range [-1.0, 1.0] meters on each axis
- **Implementation:** Direct modification of `env.data.mocap_pos`
- **Note:** An alternative XML-based implementation exists but is commented out

### B.3.5 FLOOR TEXTURE VARIATION

The floor texture factor replaces the default floor texture:

- **Source:** PNG files from a figure batch
- **Implementation:** XML texture reference modification for texture named "T_floor"
- **Error handling:** 10 attempts with different random textures to ensure successful loading

### B.4 WRAPPER CONFIGURATION

The factor wrapper is configured through a 5-character binary string parameter, where each character controls whether a specific factor is applied (1) or not (0). The factors are applied in the following order:

1. Lighting variation
2. Table texture variation
3. Camera position variation
4. Agent initial position variation
5. Floor texture variation

### B.5 OBSERVATION SPACE

The wrapper provides a rich observation space including:

- RGB images (128×128 pixels)
- Agent proprioceptive information (end-effector and finger positions)
- Full environment state

This factor wrapper enables systematic control over environment variations while maintaining compatibility with the original Meta-World API, facilitating research into factored control and domain adaptation techniques.

## C TRAINING DETAILS

**Time hyper-parameters.** Tabel 3 summarizes the key hyperparameters used during the training process, covering critical aspects such as the diffusion process, network architecture, training setup, data configuration, and inference. These parameters were carefully tuned to optimize the model's generalization performance and training stability in complex environments.

**Time efficiency.** The CPU used for the experiment is the AMD Ryzen Threadripper 3960X 24-Core Processor, and the GPU is NVIDIA GeForce RTX 3090Ti. Taking the Handle-Pull task as an example, the time taken to train for 2000 epochs in our framework is approximately 10 hours.

Table 3: Summary of key hyperparameter configurations

| Parameter Description | Parameter Name | Value |
|---|---|---|
| **Diffusion Process** | | |
| Number of diffusion timesteps | num_train_timesteps | 50 |
| Noise schedule | beta_schedule | squaredcos_cap_v2 |
| Prediction target | prediction_type | epsilon |
| **Network Architecture** | | |
| Feature dimension | feature_dim | 64 |
| U-Net decoder channels | down_dims | [256, 512, 1024] |
| Convolution kernel size | kernel_size | 5 |
| Group normalization groups | n_groups | 8 |
| Condition modulation type | condition_type | film |
| **Training Configuration** | | |
| Batch size | batch_size | 32 |
| Number of epochs | num_epochs | 2000 |
| Base learning rate | lr | 0.0001 |
| Optimizer | optimizer | AdamW |
| Weight decay | weight_decay | 0.000001 |
| Gradient accumulation steps | gradient_accumulate_every | 1 |
| EMA decay | use_ema | true |
| **Data Configuration** | | |
| Observation history steps | n_obs_steps | 2 |
| Prediction horizon | horizon | 4 |
| Action steps | n_action_steps | 4 |
| Data loading workers | num_workers | 8 |
| **Inference** | | |
| Number of denoising steps | num_inference_steps | 16 |

## D  TRAINING EVAL DETAILS

We use the Handle-Pull task as an example to illustrate our evaluation protocol. As shown in Figure 4, the evaluation employs five distinct generalization environments, corresponding to different numbers of varying factors. From left to right, these environments represent configurations with i = 1, 2, 3, 4, and 5 factors simultaneously varied within the perturbation ranges specified in Section B. For each evaluation round, we simultaneously test the model in all five environments, obtaining five separate success rates. The average of these five success rates is then used as the final evaluation result for that round.

The specific factor combinations for each evaluation environment are as follows:

- **1-factor environment:** Camera-Pos variation only

- **2-factor environment:** Camera-Pos and Lighting variations

- **3-factor environment:** Camera-Pos, Lighting and floor texture variations

- **4-factor environment:** Camera-Pos, Lighting, floor texture, and table texture variations

- **5-factor environment:** All five factors varied simultaneously (lighting, table texture, camera position, agent position, and floor texture)

This progressive evaluation scheme allows us to systematically assess the model's robustness to increasingly challenging environmental variations, from single-factor perturbations to the most complex scenario where all five factors are simultaneously altered.

# E    SINGLE ENVIRONMENT EVAL CURVES

**The following Figures 9 to 13 present the detailed training curves for each individual evaluation environment ($i = 1$ to $i = 5$), complementing the averaged results shown in the main text (Figure 5).** These results demonstrate that our FAME method consistently achieves superior performance across every individual environmental setting, not just on average. The ability to outperform all baseline methods in each specific factor combination—from single-factor variations ($i = 1$) to the most complex scenario with all five factors simultaneously perturbed ($i = 5$)—strongly validates the robustness and generalizability of our approach.

## E.1 TRAINING RESULTS WITH 1 VARYING FACTOR ($i = 1$).

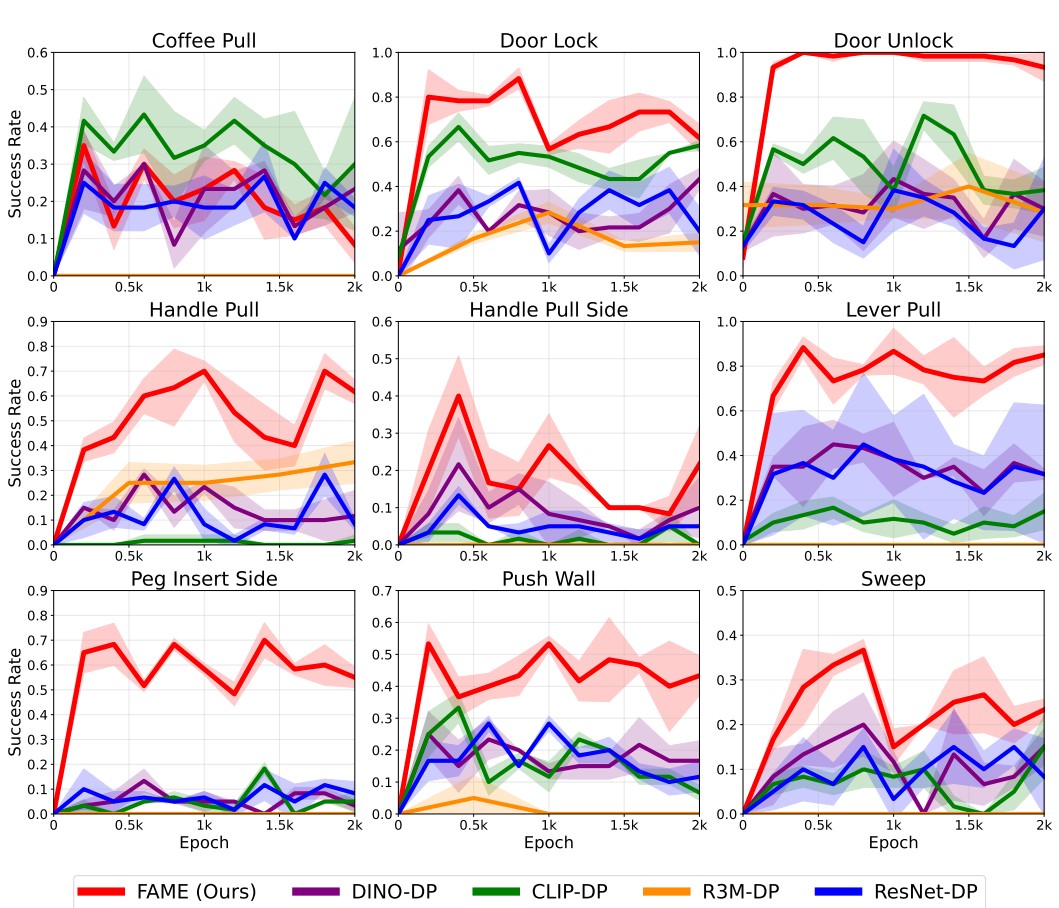

Figure 9: **Training results with 1 varying factor** ($i = 1$). The solid lines correspond to the mean and shaded regions correspond to one standard deviation over three runs. Each evaluation result is obtained from the environment with only $i = 1$ varying factor.

## E.2 TRAINING RESULTS WITH 2 VARYING FACTORS ($i = 2$).

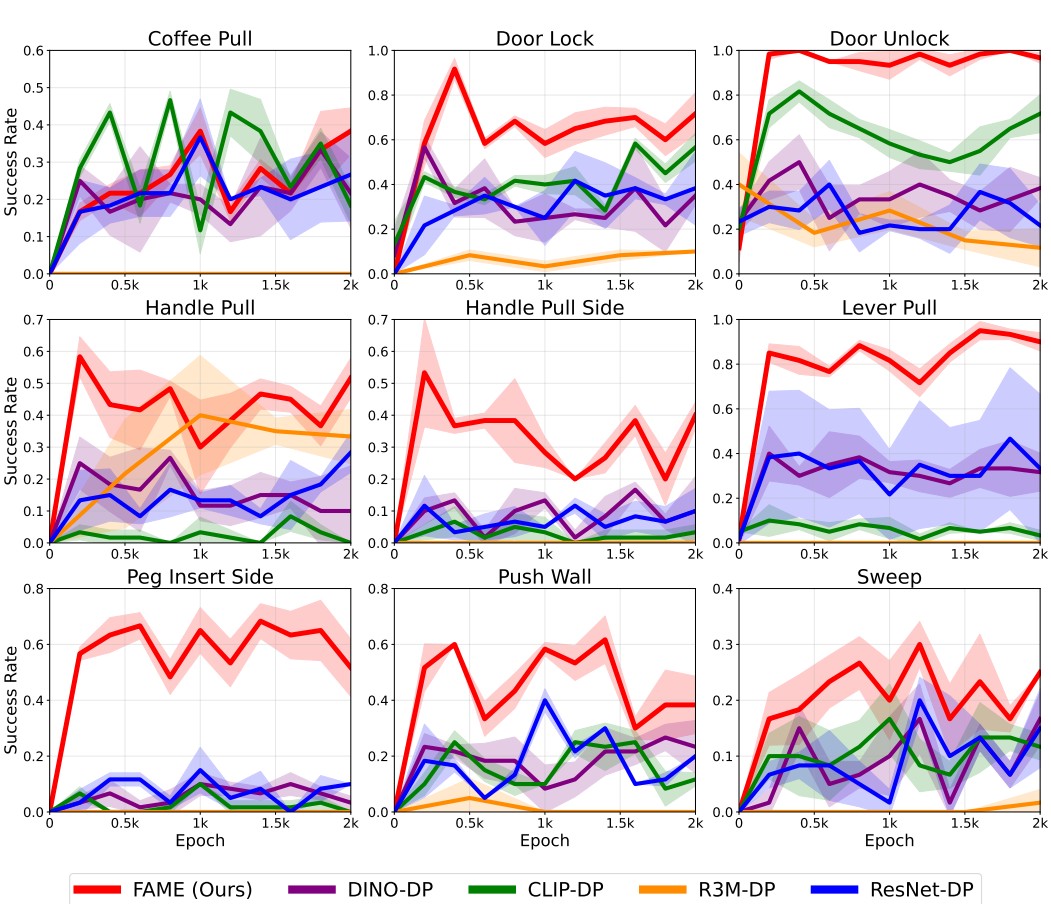

Figure 10: **Training results with 2 varying factors** ($i = 2$). The solid lines correspond to the mean and shaded regions correspond to one standard deviation over three runs. Each evaluation result is obtained from the environment with only $i = 2$ varying factor.

E.3    TRAINING RESULTS WITH 3 VARYING FACTOR ($i = 3$).

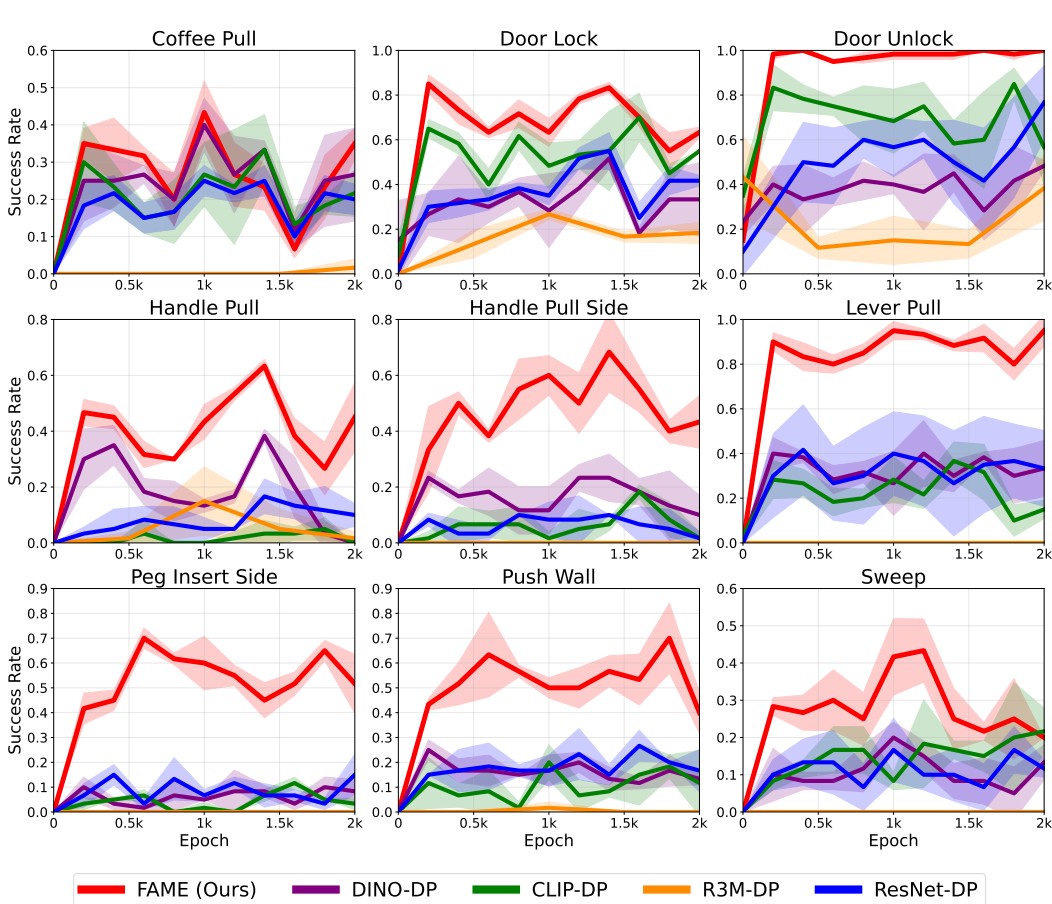

Figure 11: **Training results with 3 varying factors** ($i = 3$). The solid lines correspond to the mean and shaded regions correspond to one standard deviation over three runs. Each evaluation result is obtained from the environment with only $i = 3$ varying factor.

### E.4 TRAINING RESULTS WITH 4 VARYING FACTOR ($i = 4$).

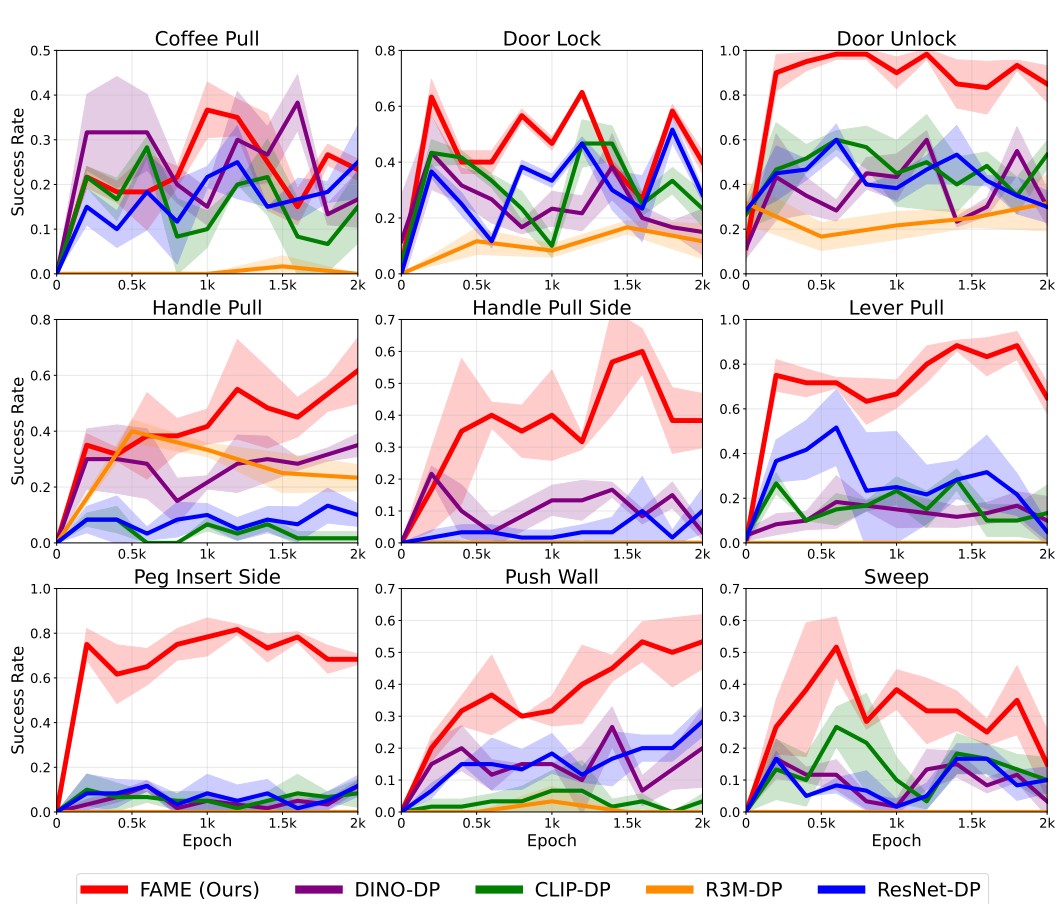

Figure 12: **Training results with 4 varying factors** ($i = 4$). The solid lines correspond to the mean and shaded regions correspond to one standard deviation over three runs. Each evaluation result is obtained from the environment with only $i = 4$ varying factor.

## E.5 TRAINING RESULTS WITH 5 VARYING FACTOR ($i = 5$).

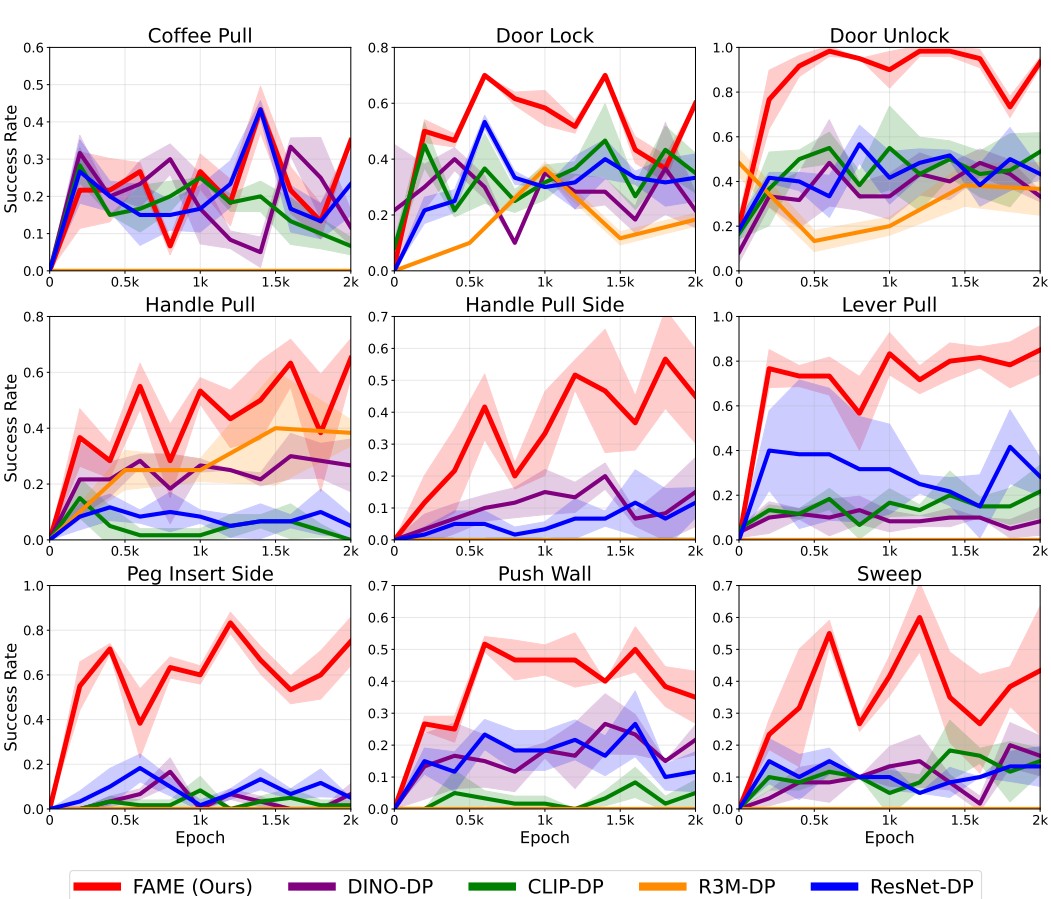

Figure 13: **Training results with 5 varying factors** ($i = 5$). The solid lines correspond to the mean and shaded regions correspond to one standard deviation over three runs. Each evaluation result is obtained from the environment with only $i = 5$ varying factor.

## ACKNOWLEDGEMENTS

We would like to express our gratitude to the AI language models that assisted in the polishing and refinement of this paper, including GPT-5, DeepSeek, and Gemini. These models provided valuable assistance in improving the clarity, coherence, and overall quality of the writing. However, all technical content, experimental results, and scientific contributions remain entirely our own work.

