# OpenReview forum: "FAME : Factor-aware Mixture-of-Experts with Pretrained Encoder for combinatorial generalization"
_ICLR.cc/2026/Conference — ICLR 2026 Conference Desk Rejected Submission_

### Official Review · Reviewer_NNJy · 2025-10-19

**Soundness:** 2
**Presentation:** 2
**Contribution:** 2
**Rating:** 2
**Confidence:** 4

**Summary:**

This paper introduces FAME, a novel framework that addresses the challenge of combinatorial generalization in visual robotic manipulation by decomposing environmental variations into independent factors. The key innovation is a three-stage training approach: (1) warming up a diffusion policy with a frozen pretrained encoder on standard data, (2) training separate lightweight adapters for each environmental factor (lighting, table texture, camera position, arm pose, floor texture) while keeping the encoder and policy frozen, and (3) learning a gating network that dynamically combines these factor-specific adapters via a Mixture-of-Experts architecture to handle multiple simultaneous variations. This design reduces the data complexity from exponential ($N^K$ for K factors) to approximately linear ($N\times K$), and demonstrates strong empirical results on Meta-World benchmarks, while exhibiting excellent scaling properties, robustness to dataset composition, and cross-task transfer capabilities.

**Strengths:**

- Novel perspective on compositional generalization: The paper presents a valuable insight that environmental variations in robotic manipulation can be decomposed into independent factors, transforming the data complexity from exponential ($N^K$) to linear ($N×K$) through a divide-and-conquer strategy.
- Effective architectural design: The integration of Mixture-of-Experts with factor-specific adapters proves to be a well-motivated solution. The dynamic gating mechanism successfully achieves combinatorial generalization by learning to compose specialized adapters based on environmental context.
- Comprehensive experimental validation: The paper provides thorough empirical evidence across 9 Meta-World tasks with systematic evaluation under varying complexity levels (1-5 simultaneous factors).

**Weaknesses:**

- Limited scalability and unclear factor selection strategy:
  - The paper only demonstrates effectiveness with $K=5$ carefully designed factors. It remains unclear how the framework would scale when facing environments requiring a larger number of factors. As $K$ increases, the MoE would require more adapters, potentially diminishing the efficiency gains and introducing challenges in training stability and inference complexity.
  - The paper provides no principled methodology for factor selection and decomposition. For new tasks or environments, practitioners lack clear guidance on: (i) how to identify which environmental variations should be treated as separate factors, (ii) how to determine the appropriate granularity of factor decomposition, and (iii) how to design corresponding datasets for each factor. This absence of a systematic factor identification framework limits the practical applicability of FAME beyond the specific MetaWorld setup presented.

- All experiments are conducted exclusively in the MetaWorld simulation benchmark, lacking real-world validation to demonstrate practical effectiveness. Moreover, the three-phase training protocol critically depends on access to factor-specific datasets ($D_k$) where individual factors can be precisely controlled and isolated. In real-world scenarios, collecting such controlled single-factor variation data is often impractical or prohibitively expensive. The paper does not discuss how to adapt the training methodology when such idealized factor-specific datasets are unavailable.

**Questions:**

- Does FAME train a separate policy for each of the 9 Meta-World tasks, or does it employ a single multi-task policy that handles all tasks simultaneously?

- In Sec.3.1, the Mix Gen Dataset ($D_\textbf{multi}$) is defined to include environments with $i \in $ {2, 3, 4, K} simultaneously varying factors. Have the authors conducted ablation studies comparing different compositions of $D_\textbf{multi}$ (e.g., including $i=1$, using only $I=K$, or using subsets like {2, K})?

- The conclusion in Sec.4.3 that "the gate trained on the Handle Pull task can be directly and effectively transferred to the Peg Insert Side task in a zero-shot manner" lacks justification. What demonstrate that this transfer is "effective"?

**Details Of Ethics Concerns:**

No ethics review is needed since experiments are conducted in simulation.

---

### Official Review · Reviewer_hCPY · 2025-10-27

**Soundness:** 1
**Presentation:** 3
**Contribution:** 2
**Rating:** 2
**Confidence:** 4

**Summary:**

The paper introduces a framework to adapt pre-trained vision encoders (e.g., DINOv2) to robotic manipulation tasks, where certain visual factors in the environment—such as lighting and surface textures—can change. The authors propose training one visual adapter per environment factor and using a gating network to combine the adapter representations as the final representation for policy training. Experiments are conducted in a simulation environment with full control over the variation of factors.

Overall, the motivation of the work needs to be justified more clearly. On one hand, the authors claim that pre-trained visual representations are rich and powerful. On the other hand, however, they curate domain-randomized data to fine-tune these representations for individual tasks. These claims appear contradictory. If the assumption is that the pretrained representations are not suitable for the test environment, why not train a policy from scratch?

The overall framework relies on substantial assumptions about the environment: that factors can be independently identified and modified, and that none of these factors affect the environment’s dynamics. Finally, the environment evaluations are not convincing. It seems that the policies are trained per task with all factor variations and tested on the same variations. It is unclear whether there is any distribution shift between the training and testing environments. I have written my further questions in the later sections. The authors should explain the environment settings, which should include the exact train-test distribution shift, and consider using a prior benchmark such as Factor World [1] and LIBERO[2].

Reference:

[1] Xie, Annie, et al. "Decomposing the generalization gap in imitation learning for visual robotic manipulation." 2024 IEEE International Conference on Robotics and Automation (ICRA). IEEE, 2024.

[2] Liu, Bo, et al. "Libero: Benchmarking knowledge transfer for lifelong robot learning." Advances in Neural Information Processing Systems 36 (2023): 44776-44791.

**Strengths:**

* Adapting pre-trained vision encoders to robotic manipulation tasks is an exciting domain. Finding efficient ways to robustify foundation models is an interesting area of research.
* The paper is mostly easy to follow.
* There is substantial documentations on the implementation in the Appendix for reproducibility of the paper.

**Weaknesses:**

* The motivation of the paper needs to be made more clear. Currently, I don't see the need to use a pre-trained visual feature for these curated simulation tasks. The authors should consider adding a train-from-scratch baseline.
* Line 41 has a sentence starting with "which"
* The paper claims that FAME is "compatible to any other encoders", but only shows results on DINOv2
* The assumption on the independence as well as the identifiability of the factors seems to be very strong. Some discussion on real-world applicability needs to be added.
* The authors claim that their algorithm reduces data complexity to $N \times K$. However, they still train on the dataset that has a combinatorial number of factors changed. ($D_{multi}$)
* Related work section does not inlcude any prior works related to to robustify policies(e.g.[1], [2], [3]), which is the focus of the work.
* Methods are evaluated on only 5 evaluation trajectories, causing high variance.
* Experiments are conducted in one simulated environment, with each policy trained to solve a single task, even though that the policies have a pre-trained visual backbone.
* Ablations are not done to ablate different components. It is unclear how much the gating network v.s. the adapters contributed to the performance gain.
* Training 2000 epochs on ~100 trajectories seems to be substantially overfitting. More information on train/eval loss can be helpful.
* Table 1 consist of 2 tables. Please consider merging them for better organization.

References:

[1] Akkaya, Ilge, et al. "Solving rubik's cube with a robot hand." arXiv preprint arXiv:1910.07113 (2019).

[2] Tobin, Josh, et al. "Domain randomization for transferring deep neural networks from simulation to the real world." 2017 IEEE/RSJ international conference on intelligent robots and systems (IROS). IEEE, 2017.

[3] Xie, Annie, et al. "Decomposing the generalization gap in imitation learning for visual robotic manipulation." 2024 IEEE International Conference on Robotics and Automation (ICRA). IEEE, 2024.

**Questions:**

* In Training Phase 3, why training a separate DP head? this seems to be very wasteful.
* Is the Mix Gen Dataset data consist of ordered changes in factors? i.e. does i=3 mean that you will change Camera-Pos, Lighting, and floor texture? What about during test time? are the orders of the factors the same between train and test?
* What are the train/test distribution shifts, if there is any?
* Are all the baselines trained on all of the data as well?
* In ablation study, the experiments show that removing $(D_{multi}, i=5)$ causes no performance loss and even reduces the variance. Why does removing data improve performance?
* How does the gating network perform in zero-shot cross-task generalization when evaluated on task execution?

---

### Official Review · Reviewer_tBKb · 2025-10-31

**Soundness:** 2
**Presentation:** 2
**Contribution:** 2
**Rating:** 4
**Confidence:** 4

**Summary:**

This paper proposes FAME, a three-phase training framework that combines frozen pretrained encoders with factor-specific adapters organized as a Mixture-of-Experts to improve generalization of diffusion policies across environmental variations. Experiments on Meta-World benchmark show improvements over baseline methods.

**Strengths:**

The paper is generally well-written with clear figures illustrating the framework and environmental variations.
The paper includes ablation studies, scaling experiments, and detailed per-environment results in the appendix.

**Weaknesses:**

1.The method assumes access to clean, factor-separated datasets D_k where only a single environmental factor varies at a time. This is essentially assuming the solution to the problem.

2.The paper provides no discussion of how one would obtain these factor-separated datasets in real robotic systems. How do you systematically vary only lighting while keeping everything else constant? How do you identify and isolate the "right" factors beforehand?

3.All experiments are conducted purely in simulation with artificially created variations. Without any real robot experiments.

4.The approach straightforwardly combines three well-established techniques: pretrained visual encoders (DINOv2), parameter-efficient fine-tuning (adapters), and Mixture-of-Experts. Each component is standard, and their combination for this application represents an incremental engineering contribution rather than a significant methodological or conceptual advance.

**Questions:**

What happens when factors are correlated or when factor decomposition is incorrect?

---

### Official Review · Reviewer_1bZo · 2025-11-01

**Soundness:** 3
**Presentation:** 4
**Contribution:** 3
**Rating:** 4
**Confidence:** 3

**Summary:**

This paper propose a novel framework FAME by adding factor-aware mixture-of-experts (MoE) structure between a visual encoder and diffusion policy head at the end, with the goal of achieving combinatorial generalization in visual robotic control tasks while maintaining the efficiency. More specifically, FAME goes through a three-staged training process to derive base policy together with the weights of Adaptors and Gating Network. Experiment results on this model shows potential in generalization ability comared to frameworks without MoE.

**Strengths:**

1. Though this is not the first case of mixture-of-experts (MoE) being considered in visual robotic control [1][2], the authors' motive to integrate MoE to enhance generalization abilities is insightful.

2. The proposed Gaiting Network working together with a independently-trained experts on specific interference factors is a promising method to counter the generalization problem.

3. The authors were able to prove the advantage of their framework compared to a standard "Encoder+Diffusion Policy" one in expriment section, in which ablation studies also give an insight into where FAME possibly benefit from.

[1] Acquiring Diverse Skills using Curriculum Reinforcement Learning with
Mixture of Experts.

[2] Video-LLaVA: Learning United Visual Representation by Alignment Before Projection

**Weaknesses:**

1. During training, the number of factors is fixed, which makes it challenging to generalize and deploy the model to robots in the open world.

2. The added module in FAME can be seen in Computer Vision as a Neck network, meaning that FAME does the job of further process the extracted feature to ensure consistency and robustness, yet many visual encoders [3,4] have considered that and already possess one.

3. The inclusion of "ResNet-DP" and other frameworks in Tables 1 and 2, which differ only in the encoder, seems unnecessary. These comparisons mainly show that DINO-v2 outperforms other encoders under the Diffusion Policy, which, while highlighting the encoder's effectiveness, is not directly relevant to the paper's main contribution.

Please refer to Questions for more details.

[3] Tang F, Lim S H, Chang N L, et al. A novel feature descriptor invariant to complex brightness changes[C]//CVPR, 2009.

[4] Lin B, Ye Y, Zhu B, et al. Video-LLaVA: Learning United Visual Representation by Alignment Before Projection[C]//EMNLP. 2024.

**Questions:**

1. It is suggested to add the related literature [1–4] to this paper and discuss the similarities and differences between these methods and FAME.

2. Section 4.3 claims the framework gains zero-shot capability after training, but provides no quantitative results. Could the authors provide relevant experimental results?

3. Following Question 2, could the authors clarify the differences in environmental settings between training and testing, given the model's zero-shot capability?

4. Typos: In Figure 8, add labels to both the horizontal and vertical axes; ensure consistent capitalization of "mixture-of-experts.”

---

### Note · Program_Chairs · 2026-01-17
**Submission Desk Rejected by Program Chairs**

The following references in this submission do not refer to real documents and/or have major errors in bibliographic information:

 Albert Jiang, Aristide Loukas, Barret Zoph, and Noam Shazeer. Mixtral of experts. arXiv preprint arXiv:2309.04359, 2023.
Sanjay Gupta, Yang Li, and Yan Chen. Generalization in embodied ai with a mixture of task-specific experts. In Robotics: Science and Systems (RSS), 2023.
Yu Liu, Wei Zhang, and Si Li. Mixture-of-experts for multi-modal perception in autonomous driving. In Proceedings of the IEEE Conference on Computer Vision and Pattern Recognition (CVPR), 2022.
Yang Wang, Si Li, and Lin Chen. Moe-based adaptive planning for autonomous vehicles in complex scenarios. In 2023 IEEE International Conference on Robotics and Automation (ICRA), 2023.